# Denial of Justification for Vaccination: Its Multiple Related Variables and Impacts on Intention to Get Vaccinated against COVID-19

**DOI:** 10.3390/vaccines9080822

**Published:** 2021-07-25

**Authors:** Yen-Ju Lin, Wen-Jiun Chou, Yu-Ping Chang, Cheng-Fang Yen

**Affiliations:** 1Department of Psychiatry, School of Medicine, College of Medicine, Kaohsiung Medical University, Kaohsiung 80708, Taiwan; 1040457@kmuh.org.tw; 2Department of Psychiatry, Kaohsiung Medical University Hospital, Kaohsiung 80708, Taiwan; 3School of Medicine, Chang Gung University, Taoyuan 33302, Taiwan; wjchou@cgmh.org.tw; 4Department of Child and Adolescent Psychiatry, Chang Gung Memorial Hospital, Kaohsiung Medical Center, Kaohsiung 83301, Taiwan; 5School of Nursing, The State University of New York, University at Buffalo, New York, NY 14214-3079, USA; yc73@buffalo.edu

**Keywords:** COVID-19, intention, perceived risk, vaccination, vaccine refusal

## Abstract

The aims of the present study were (1) to identify the variables related to denying the justification for vaccination during the coronavirus disease 2019 (COVID-19) pandemic in Taiwan and (2) to examine the associations of such denial with perceived risk of COVID-19 and the extrinsic and intrinsic intentions to get vaccinated against COVID-19. We recruited 1047 participants by using a Facebook advertisement. We investigated whether the participants denied justification for vaccination as well as their sociodemographic characteristics, mental health status, sources of information about COVID-19 vaccination, perceived risk of COVID-19, and extrinsic and intrinsic intentions to get vaccinated against COVID-19. The results indicated that 20.0% of the participants denied justification for vaccination. Participants who were older, had an educational level below college, were not health care workers, were in poor general mental health state, or did not obtain information about COVID-19 vaccination from the Internet were more likely to deny justification for vaccination. Denial was negatively associated with both extrinsic and intrinsic intentions to get vaccinated against COVID-19 but not associated with the perceived risk of COVID-19. Multiple variables related to denying the justification for vaccination; the denial was negatively associated with the intention to get vaccinated against COVID-19.

## 1. Introduction

### 1.1. Importance of Vaccination against Coronavirus Disease 2019

The coronavirus disease 2019 (COVID-19) has caused significant harm to people worldwide [1,2,3,4,5,6,7,8]. Vaccines have been developed in a short period and have led people to have great expectations for them to suppress the spread of COVID-19 [9,10]. Researchers estimated that the herd immunity threshold for COVID-19 requires that approximately 67% of a population be vaccinated [11]. However, hesitancy to get vaccinated against COVID-19 is prevalent worldwide [12,13,14,15]. Vaccine hesitancy may render it a challenge to reach the herd immunity threshold for COVID-19 through vaccination [12]. Examining people’s beliefs about vaccination against COVID-19 and related variables is important for developing intervention programs to stop the spread of COVID-19.

### 1.2. Vaccine Hesitancy: A Multifactorial Problem

The World Health Organization (WHO) working group defined vaccine hesitancy as “a delay in acceptance or refusal of vaccines despite availability of vaccination services” [16]. Vaccine hesitancy is the result of the interaction of multiple cognitive and behavioral variables. For example, the 3Cs model lists the variables influencing vaccine hesitancy as: confidence (low trust in either the vaccine or provider); complacency (no perceived need or value for the vaccine), and convenience (insufficient access to vaccines) [17]. The 5As model proposes five fundamental causes of vaccine hesitancy: challenges to access (reach to or be reached by recommended vaccines); affordability (regarding the financial and nonfinancial costs of vaccination); awareness (knowledge of the need for recommended vaccines and their objective benefits and risks); acceptance, and activation (nudging individuals toward vaccination) [18]. Therefore, vaccine hesitancy should be understood as multifactorial.

### 1.3. Denial of Justification for Vaccination

People may reject vaccination by denying the need for, value of, and justification for vaccination. Such individuals may deny the justification for vaccination on the basis of religious [19] or philosophical objections, such as a desire to live a natural life [20]. Deniers may also object to vaccines for safety reasons [21], such as the concern regarding a vaccine–autism connection, which was based on unethical medical practices and fraudulent science [22]. Difficulties in accessing or affording a vaccine are not the main concerns of individuals who deny the justification for vaccination. Therefore, such denial is a unique cause and form of vaccine hesitancy and warrants specific intervention. As it may impede efforts to reach the herd immunity threshold for COVID-19 through vaccination, further study is needed to examine the variables related to denial of vaccination justification and its relationships with risk perception and the intention to get vaccinated against COVID-19.

### 1.4. Study Aims and Hypotheses

The aims of the present study were to identify the variables related to denial of the justification for vaccination in Taiwan and to examine the associations of such denial with the perceived risk of COVID-19 and extrinsic and intrinsic intentions to get vaccinated.

We had two hypotheses. First, according to the 5As model, knowledge of the need for vaccination is one of fundamental causes of vaccine hesitancy [18]. Therefore, we hypothesized that variables that may influence the individuals’ receiving and absorbing knowledge about vaccination against COVID-19 such as age, educational level, health care workers, mental health state, and sources of information about COVID-19 would be related to denying justification for vaccination. Second, according to the 3Cs model, perceived need for vaccination is one of the variables influencing vaccine hesitancy [17]. Moreover, according to the health belief model [23,24], risk perception is the most crucial factor in promoting self-protective behaviors against respiratory infectious diseases [25]. Therefore, we hypothesized that denial of justification for vaccination would be negatively associated with the perceived risk of COVID-19 and extrinsic and intrinsic intentions to get vaccinated.

## 2. Methods

### 2.1. Participants

A previous study has described the study design and the method of recruiting participants [26]. The study period was from 15 October 2020 to 21 December 2020. We recruited 1047 participants who were aged ≥20 years and lived in Taiwan at the time of the Facebook advertisement. As of 21 December 2020, 627 patients had contracted COVID-19, in Taiwan, and 7 patients had died [27]. No vaccine against COVID-19 was available in Taiwan during the study period. The Institutional Review Board of Kaohsiung Medical University Hospital approved this study (KMUHIRB-EXEMPT(I) 20200019).

### 2.2. Measures

#### 2.2.1. Denial of Justification for Vaccination

We asked the participants their level of agreement with the following statement: “Humans should not be vaccinated at all.” We scored their answers as 0 (*strongly disagree*), 1 (*disagree*), 2 (*agree*), or 3 (*strongly agree*). Scores of 2 or 3 and scores of 0 or 1 were considered to reflect denial and no denial of justification for vaccination, respectively. We also asked participants their experience with influenza vaccination (received or never received vaccination). The result of a χ^2^ test revealed that participants who denied the justification for vaccination were more likely to have never received vaccination against influenza than were those who did not deny the justification (χ^2^ = 17.920; *p* < 0.001). The results support the validity of the question regarding denial of vaccination justification used in the present study.

#### 2.2.2. Brief Symptom Rating Scale

General mental health state was assessed by using the Brief Symptom Rating Scale (BSRS-5), which contains five items measuring, self-reported anxiety, depression, hostility, inferiority, and insomnia in the past week. Each item is rated on a 5-point scale from 0 (*not at all*) to 4 (*extremely*). The Cronbach α value in this study was 0.897. Participants whose total BSRS-5 score was ≥10 were classified as being in poor general mental health state [28].

#### 2.2.3. Sources of Information about COVID-19 Vaccine

We asked the participants whether they had obtained COVID-19 vaccination information from the Internet (e.g., Facebook, Twitter, blogs, or news sites), traditional media (e.g., newspapers, television, or radio), or friends and family members [29]. Table 1 presents the questions and scoring thereof. We divided the participants into those who had and had not obtained COVID-19 vaccination information from those sources.

#### 2.2.4. Extrinsic Intention to Receive COVID-19 Vaccination

The extrinsic intention to receive COVID-19 vaccination was assessed using one item. Table 1 presents the items and scoring thereof. A higher score indicated a greater extrinsic intention to get vaccinated [29].

#### 2.2.5. Intrinsic Intention Receive COVID-19 Vaccination

We used the Drivers of COVID-19 Vaccination Acceptance Scale (DrVac-COVID19S) to measure self-reported intrinsic intention to receive COVID-19 vaccination [30]. The DrVac-COVID19S measures four cognitive traits regarding the intention to receive COVID-19 vaccination: values (three items regarding the degree of care for the purpose of COVID-19 vaccination), impacts (three items regarding belief in COVID-19 vaccination to prevent COVID-19 infection), knowledge (three items related regarding knowledge about COVID-19 vaccination), and autonomy (three items related to confidence in control over receiving the COVID-19 vaccination). Table 1 presents the items and scoring thereof. A higher total score indicated greater intrinsic intention to get vaccinated against COVID-19 [30]. The Cronbach α in this study was 0.867.

#### 2.2.6. Perceived Risk of COVID-19

A five item questionnaire was used to measure the perceived risk of COVID-19 [31]. The five items assessed worries about developing flu-like symptoms, worries about the possibility of contracting COVID-19, worries about COVID-19, the perceived likelihood of contracting COVID-19, and of the perceived relative likelihood of contracting COVID-19 as compared with individuals outside the family. Table 1 presents the items and scoring thereof. A higher total score indicated greater perceived risk. The Cronbach α was 0.704.

#### 2.2.7. Sociodemographic Characteristics

Information about the participants’ sex (female or male), age, educational level (below college or college or above), and occupation (health care workers or other occupation) was collected.

### 2.3. Data Analysis

We performed data analysis using IBM SPSS Statistics version 24.0 software (IBM Corp., Armonk, NY, USA). The sex, age, educational level, occupational classification, general mental health state, and sources of information about COVID-19 vaccination of participants who denied and did not deny the justification for vaccination were compared using χ^2^ and *t* tests. The significant variables were entered into logistic regression analysis to examine their associations with denial of the justification for vaccination.

The levels of risk perceived of COVID-19 and the extrinsic and intrinsic intentions to get vaccinated of the participants who denied and did not deny the justification for vaccination were compared using *t* tests. The associations of denial with perceived risk and extrinsic and intrinsic intentions to get vaccinated were further examined through multiple regression controlling for the effects of sociodemographic characteristics. A two-tailed *p* value of <0.05 indicated statistical significance.

## 3. Results

In total, 209 (20.0%) participants were classified as denying the justification for vaccination, and 838 (80.0%) were classified as not denying the justification. Table 2 presents the results of our comparison of the demographic characteristics, mental health status, and sources of information about COVID-19 vaccination of the participants who denied and did not deny the justification for vaccination. Participants denying the justification were more likely to be female, to have an educational level below college, to be in poor general mental health state, and to have obtained information about COVID-19 vaccination from friends and family members and less likely to be health care workers and to have obtained information about COVID-19 vaccination from the Internet. Participants denying the justification were also older than those not denying the justification. The participants denying and not denying the justification did not differ significantly in obtaining COVID-19 vaccination information from traditional media.

The significant variables in the χ^2^ and *t* tests were entered into logistic regression analysis to examine their associations with denial of the justification for vaccination (Table 3). Participants who were older, had an educational level below college, were not health care workers, were in poor general mental health state, or did not obtain information about COVID-19 vaccination from the Internet were more likely to deny the justification for vaccination. Sex and obtaining information from friends and families were not significantly associated with denial of justification for vaccination.

Table 4 presents the results of the *t* tests comparing the levels of perceived risk and extrinsic and intrinsic intentions to get vaccinated against COVID-19 of the participants denying and not denying the justification for vaccination. Perceived risk did not differ significantly, but denying participants had lower levels of extrinsic and intrinsic intentions to get vaccinated against COVID-19 than did those not denying the justification.

The associations between denial of the justification for vaccination and extrinsic and intrinsic intentions to get vaccinated against COVID-19 were further examined using multiple regression (Table 5). With the effects of sociodemographic characteristics controlled for, denial was negatively associated with both extrinsic and intrinsic intentions to get vaccinated against COVID-19.

## 4. Discussion

We revealed that 20.0% of our participants denied the justification for vaccination. Multiple variables, including age, educational level, occupation, general mental health state, and sources of information about COVID-19 vaccination, were related to denial of justification for vaccination. In addition, denial was negatively associated with both extrinsic and intrinsic intentions to get vaccinated against COVID-19. 

Before discussion, the limitation of the present study resulting from the method of recruiting participants should be addressed. Although recruiting participants from Facebook is a practical method during the pandemic, the participants might not be representative of the general population [32]. People are not equally motivated to use Facebook [33]. Moreover, a 2018 analysis found that 68.4% of active Facebook users in Taiwan were aged between 18 and 44 [34].

### 4.1. Association of Denial of Justification with Intention to Receive COVID-19 Vaccination

One-fifth of the participants agreed or strongly agreed that humans should not be vaccinated; such a negative attitude toward vaccination was associated with a weak intention to get vaccinated against COVID-19. Denial of justification for vaccination reflects a strong objection to vaccination and disregard for the scientific evidence of its safety and effectiveness. Although the WHO has globally promoted the importance of vaccination against COVID-19 [35], such a high prevalence of vaccine denial could render achieving herd immunity to COVID-19 through vaccination exceedingly difficult. Interventions aimed at attenuating negative attitudes toward vaccination against COVID-19 are urgently needed. A study on the effectiveness of interventions for parents refusing vaccines proposed educational models for increasing acceptance of the necessity of vaccination and correcting false information about vaccination; however, the effectiveness of such intervention warrants further study [36]. Health professionals should address denial of the justification for vaccination by respectfully listening to deniers’ concerns and discussing the risks of not receiving vaccination [37].

### 4.2. Variables Related to Denial of Justification for Vaccination

Older age, an educational level below college, not working in health care, poor general mental health state, and not obtaining information about COVID-19 vaccination from the Internet were significantly associated with denying the justification for vaccination. Older age, a low education level, and not working in health care may limit people’s opportunities to receive updated knowledge about how vaccination protects people from contracting infectious diseases; therefore, their existing attitude of denying the justification for vaccination may remain unchanged. Research has demonstrated that health workers had a higher level of risk perception of COVID-19 [31], adopted more protective behaviors against COVID-19 [31], and had higher motivation to receive COVID-19 vaccination than the public [26]. The results of recent studies echoed that of the present study. Denial was also associated with not obtaining information about COVID-19 vaccination from the Internet. Without the Internet, delivering accurate information about vaccination and the population’s positive experiences with it to deniers is challenging. Enhancing positive attitudes toward vaccination and increasing the rate of COVID-19 vaccination remain challenging to governments worldwide.

Poor general mental health state was associated with denial of the justification for vaccination. Several mechanisms may account for this association. First, poor general mental health state may limit an individual’s ability to access adequate and timely information about vaccination. Second, denial of justification for vaccination may indicate an individual’s mistrust of modern medicine, which may also delay the treatment of their own mental health problems. Third, neurocognitive function may be a basis of both mental health and learning [38]. Because mental health problems are prevalent during the COVID-19 pandemic [39] and poor general mental health state was associated with denial of the justification for vaccination, improving individuals’ attitudes toward vaccination and their mental health is simultaneously crucial.

### 4.3. Limitations

The present study has some limitations. In addition to the limitation resulting from recruiting participants from Facebook, the cross-sectional study design limits the inference of temporal relationships among the variables. Moreover, we did not investigate each participant’s reasons for denial. Determining the reasons for denying the justification for vaccination would benefit the development of intervention programs. The first batch of 117,000 COVID-19 vaccine doses manufactured by AstraZeneca arrived in Taiwan on 3 March 2021. [40]. Therefore, no vaccine against COVID-19 was available in Taiwan during the study period. Although information for the progression of design and clinical trials of COVID-19 vaccines authorized for emergency use is available in Taiwan, participants of this study might not have obtained enough knowledge to change their existed attitude toward vaccination. We applied the BSRS-5 to measure the general mental health state of participants in the past one week. We did not assess the mental health of participants before the COVID-19 pandemic or during a longer period.

## 5. Conclusions

One-fifth of our participants denied the justification for COVID-19 vaccination. Furthermore, multiple variables were related to denial and the extrinsic and intrinsic intentions to get vaccinated. On the basis of the results, we suggest that deniers of justification for vaccination require specific intervention programs to increase their intention to get vaccinated against COVID-19. Age, educational level, occupation, general mental health state, and sources of information should be considered in developing the intervention programs.

## Figures and Tables

**Table 1 vaccines-09-00822-t001:** Items on Sources of Information Concerning COVID-19 Vaccination, Extrinsic and Intrinsic Intentions to get Vaccinated Against COVID-19, and Perceived Risk of COVID-19.

Measures	Items	Response Scale
Sources of information concerning COVID-19 vaccination	Do you obtain COVID-19 vaccination information from (1) Internet media (e.g., Facebook, Twitter, blogs, and Internet news); (2) traditional media (e.g., newspapers, television, and radio broadcasting); (3) friends; and (4) family members?	0 = no 1 = yes
Explicit intention to get vaccinated against COVID-19	Please rate your current willingness to receive a COVID-19 vaccine:	1 (very low) to 10 (very high)
Intrinsic intention to get vaccinated against COVID-19 (Drivers of COVID-19 Vaccination Acceptance Scale)	1. Vaccination is a very effective way to protect me against COVID-19.	1 = strongly disagree, 2 = disagree, 3 = slightly disagree, 4 = neither disagree nor agree, 5 = slightly agree, 6 = agree, 7 = strongly agree
	2. I know very well how vaccination protects me from COVID-19.	
	3. It is important that I get the COVID-19 jab.	
	4. Vaccination greatly reduces my risk of catching COVID-19.	
	5. I understand how the flu jab helps my body fight the COVID-19 virus.	
	6. The COVID-19 jab plays an important role in protecting my life and that of others.	
	7.* I feel under pressure to get the COVID-19 jab.	
	8. The contribution of the COVID-19 jab to my health and well-being is very important.	
	9. I can choose whether to get a COVID-19 jab or not.	
	10.* How the COVID-19 jab works to protect my health is a mystery to me.	
	11.* I get the COVID-19 jab only because I am required to do so.	
	12. Getting the COVID-19 jab has a positive influence on my health.	
Risk perception of COVID-19	Item 1: If you were to develop flu-like symptoms tomorrow, would you worry?	1 = not at all worried, 2 = worried less than normal, 3 = about the same, 4 = worried more than normal, 5 = extremely worried
	Item 2: In the past one week, have you ever worried about catching COVID-19?	1 = no, never think about it, 2 = think about it but it didn’t worry me, 3 = worried me a bit, 4 = worried me a lot, 5 = worried about it all the time
	Item 3: Please rate the current level of your worry towards COVID-19:	Score ranged from 1–10 (1 = very mild, 10 = very severe)
	Item 4: How likely do you think it is that you will contract COVID-19 over the next 1 month?	1 = never, 2 = very unlikely, 3 = unlikely, 4 = evens, 5 = likely, 6 = very likely, 7 = certain
	Item 5: What do you think are your chances of getting COVID-19 over the next 1 month compared to others outside your family?	1 = not at all, 2 = much less, 3 = less, 4 = evens, 5 = more, 6 = much more, 7 = certain

Note: COVID-19: Coronavirus disease 2019. *: reverse-coded

**Table 2 vaccines-09-00822-t002:** Demographic Characteristics, General Mental Health State, and Sources of Information About COVID-19 Vaccination of Participants Denying and not Denying Justification for Vaccination (N = 1047).

Variables	Denial of the Justification for Vaccination			
	No (*N* = 838)	Yes (*N* = 209)	χ^2^ or *t*	*p*
Sex, *n* (%)				
Male	357 (42.6)	73 (34.9)	−4.069	0.044
Female	481 (57.4)	136 (65.1)		
Age, mean (*SD*)	34.8 (9.1)	38.5 (10.9)	−5.028	<0.001
Educational level, *n* (%)				
Below college	64 (7.6)	46 (22.0)	36.750	<0.001
College or above	774 (92.4)	163 (78.0)		
Health care workers, *n* (%)				
No	597 (71.2)	171 (81.8)	9.574	0.002
Yes	241 (28.8)	38 (18.2)		
Poor general mental health state, *n* (%)				
No	719 (85.8)	164 (78.5)	6.805	0.009
Yes	119 (14.2)	45 (21.5)		
Sources of information about COVID-19 vaccine, *n* (%)				
Internet				
No	159 (19.0)	57 (27.3)	7.036	0.008
Yes	679 (81.0)	152 (72.7)		
Traditional media				
No	250 (29.8)	59 (28.2)	0.207	0.649
Yes	588 (70.2)	150 (71.8)		
Friends				
No	592 (70.6)	131 (62.7)	4.966	0.026
Yes	246 (29.4)	78 (37.3)		
Family members				
No	615 (73.4)	136 (65.1)	5.706	0.017
Yes	223 (26.6)	73 (34.9)		

Note: COVID-19: Coronavirus disease 2019.

**Table 3 vaccines-09-00822-t003:** Associations of Sociodemographic Characteristics, General Mental Health State, and Sources of Information About COVID-19 Vaccination with Denial of Justification for Vaccination in Logistic Regression Analysis.

Variables	Denial of the Justification for Vaccination ^a^
	OR (95% CI)
Female ^b^	13.73 (0.985-1.912)
Age	1.042 (1.024-1.060) ***
Educational degree of college or above ^c^	0.369 (0.235-0.578) ***
Health care workers ^d^	0.559 (0.373-0.837) **
Poor general mental health state ^e^	2.024 (1.333-3.074) **
Information from the Internet ^f^	0.552 (0.374-0.814) **
Information from friends ^f^	1.481 (0.946-2.320)
Information from families ^f^	1.339 (0.846-2.120)

Note: COVID-19: Coronavirus disease 2019. ^a^: dependent variable; ^b^: male sex as the reference; ^c^: educational degree below college as the reference; ^d^: occupation outside health care as the reference; ^e^: good general mental health state as the reference; ^f^: not obtaining information from this source as the reference. **: *p* < 0.01; ***: *p* < 0.001.

**Table 4 vaccines-09-00822-t004:** Comparison of Perceived Risk and Extrinsic and Intrinsic Intentions to get Vaccinated Against COVID-19 of Participants Denying and not Denying Justification for Vaccination (N = 1047).

Variables	Denial of the Justification for Vaccination			
	No (*N* = 838) Mean (*SD*)	Yes (*N* = 209) Mean (*SD*)	*t*	*p*
Risk perception	17.5 (5.3)	17.9 (6.1)	−1.098	0.272
Explicit intention	6.7 (2.5)	5.7 (2.8)	5.590	<0.001
Intrinsic intention	62.9 (11.2)	58.0 (10.9)	5.194	<0.001

Note: COVID-19: Coronavirus disease 2019.

**Table 5 vaccines-09-00822-t005:** Associations Between Denial of Justification for Vaccination and Sociodemographic Characteristics and Extrinsic and Intrinsic Intentions to get Vaccinated Against COVID-19 in Multiple Regression Analysis.

Variables	Explicit Intention ^a^			Intrinsic Intention ^a^		
	*B*	*SE*	*p*	*B*	*SE*	*p*
Vaccine refusal	−0.997	0.204	<0.001	−4.696	0.874	<0.001
Sex	0.625	0.161	<0.001	4.015	0.690	<0.001
Age	−0.004	0.009	0.601	−0.033	0.037	0.371
Education level	−0.309	0.269	0.251	−1.511	1.152	0.190
Health care workers	0.147	0.184	0.425	−0.940	0.787	0.233

Note: COVID-19: Coronavirus disease 2019. ^a^: dependent variable.

## Data Availability

The data are available on reasonable request to the corresponding authors.

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
