# Peer review of "Denial of Justification for Vaccination: Its Multiple Related Variables and Impacts on Intention to Get Vaccinated against COVID-19"

_vaccines, 2021, doi:10.3390/vaccines9080822_

Round 1
Reviewer 1 Report
The authors analyze a sample of 1,047 people from Taiwan recruited through facebook, 1) to identify factors related to denying the justification for vaccination during the current pandemic and 2) study the associations between the denyal with perceived risk of covid-10 and intentions to get vaccinated as indicated in the title.
The topic is relevant since the individual decision in this respect has a direct effect also at the societal level, and this is crucial for the success of vaccination strategy against the pandemic. Therefore, understanding the multifactorial reasons behind hesitancy, which have been dissected by different explicative models, is important.
In their hypothesis, the authors should mention how their hypothesis/ data fits with these models.
The work is an extension/complementary of the previous entitled Lin, Y.-J.; Yen, C.-F.; Chang, Y.-P.; Wang, P.-W. Comparisons of motivation to receive COVID-19 vaccination and related factors between frontline physicians and nurses and the public in Taiwan: Applying the Extended Protection Motivation Theory. Vaccines. 2021, 9, 528. doi:10.3390/vaccines9050528.
The authors should discuss a little about the previous published work as it may enrich the discussion.
During the study period, no vaccine was available in Taiwan. Since the lack of information is one of the factors, the authors should discuss how it could affect the results.
The title should be informative, a kind of conclusion. As it is stated now, it refers to the aims, not to the results.
Females were more likely to deny the justification of vaccination, but this finding is not reflected in the abstract, and it is not mentioned after that. Please, clarify. If so, the sex bias is important and results should be segregated by sex.
Table 3. Usually, the sex of reference is male, so it may be confusing when interpreting the data in the tables.
The authors refer to 'poor mental health' but since the tool used was for 'general' mental health, they should better refer to 'poor general mental health'.
The BSRS-5 scale was applied, referring to the past 1 week in a study done during the pandemic. Therefore, the results of general mental health may reflect the 'state' of individuals in the current situation but may not necessarily imply that they had a trait or 'poor general mental health' before the pandemic, as the question only refers to the past week, but there's no reference to previous /longer periods. This interpretation must be included in the discussion.
Minor:
Line 142 The sex, age, educational… 'The'
In all the tables, Covid-19, coronavirus disease 2019.?
Line 198. 4. Discussion
Author Response
Thank you so much for your comments. As discussed below, we have revised our manuscript based on your suggestions. Please let me know if we need to provide anything else regarding this revision.
Comment 1
The topic is relevant since the individual decision in this respect has a direct effect also at the societal level, and this is crucial for the success of vaccination strategy against the pandemic. Therefore, understanding the multifactorial reasons behind hesitancy, which have been dissected by different explicative models, is important. In their hypothesis, the authors should mention how their hypothesis/ data fits with these models.
Response
Thank you for your comment. We rewrote the paragraph of hypothesis to integrate explicative models into our study hypotheses as below. Please refer to line 75-86.
“We had two hypotheses. First, according to the 5As model, knowledge of the need for vaccination is one of fundamental causes of vaccine hesitancy [18]. Therefore, we hypothesized that factors that may influence the individuals’ receiving and absorbing knowledge about vaccination against COVID-19 such as age, educational level, health care workers, mental health state, and sources of information about COVID-19 would be related to denying justification for vaccination. Second, according to the 3Cs model, perceived need for vaccination is one of the factors influencing vaccine hesitancy [17]. Moreover, according to the health belief model [23,24], risk perception is the most crucial factor in promoting self-protective behaviors against respiratory infectious diseases [25]. Therefore, we hypothesized that denial of justification for vaccination would be negatively associated with the perceived risk of COVID-19 and extrinsic and intrinsic intentions to get vaccinated.”
Comment 2
The work is an extension/complementary of the previous entitled Lin, Y.-J.; Yen, C.-F.; Chang, Y.-P.; Wang, P.-W. Comparisons of motivation to receive COVID-19 vaccination and related factors between frontline physicians and nurses and the public in Taiwan: Applying the Extended Protection Motivation Theory. Vaccines. 2021, 9, 528. doi:10.3390/vaccines9050528. The authors should discuss a little about the previous published work as it may enrich the discussion.
Response
Thank you for your suggestion. We added the results of two previous studies as below into Discussion section. Please refer to line 245-248.
“Research has demonstrated that health workers had a higher level of risk perception of COVID-19 [28], adopted more protective behaviors against COVID-19 [31], and had higher motivation to receive COVID-19 vaccination than the public [26]. The results of present studies echoed that of the present study.”
Comment 3
During the study period, no vaccine was available in Taiwan. Since the lack of information is one of the factors, the authors should discuss how it could affect the results.
Response
Thank you for your suggestion. We listed it as one of limitations in this study as below. Please refer to line 269-275.
“The first batch of 117,000 COVID-19 vaccine doses manufactured by AstraZeneca arrived in Taiwan on March 3, 2021. [40]. Therefore, no vaccine against COVID-19 was available in Taiwan during the study period. Although information for the progression of design and clinical trials of COVID-19 vaccines authorized for emergency use is available in Taiwan, participants of this study might have no enough knowledge obtained to change their existed attitude toward vaccination.”
Comment 4
The title should be informative, a kind of conclusion. As it is stated now, it refers to the aims, not to the results.
Response
We revised the title into “Denial of Justification for Vaccination: Its Multiple Related Factors and Impacts on Intention to Get Vaccinated Against COVID-19”. Please refer to line 2-4.
Comment 5
Females were more likely to deny the justification of vaccination, but this finding is not reflected in the abstract, and it is not mentioned after that. Please, clarify. If so, the sex bias is important and results should be segregated by sex.
Response
Although the result of chi-square test indicated that females were more likely to deny the justification of vaccination, sex difference was not significant in multiple logistic regression analysis. We added the description as below into the revised manuscript. Please refer to line 184-185.
“Sex and obtaining information from friends and families were not significantly associated with denial of justification for vaccination.”
Comment 6
Table 3. Usually, the sex of reference is male, so it may be confusing when interpreting the data in the tables.
Response
We changed the reference of sex into male and revised the content of Table 3.
Comment 7
The authors refer to 'poor mental health' but since the tool used was for 'general' mental health, they should better refer to 'poor general mental health'.
Response
We revised the term into “poor general mental health state” thorough the revised manuscript.
Comment 8
The BSRS-5 scale was applied, referring to the past 1 week in a study done during the pandemic. Therefore, the results of general mental health may reflect the 'state' of individuals in the current situation but may not necessarily imply that they had a trait or 'poor general mental health' before the pandemic, as the question only refers to the past week, but there's no reference to previous /longer periods. This interpretation must be included in the discussion.
Response
We revised the term into “poor general mental health state” thorough the revised manuscript. we also added discussion as below. Please refer to line 275-278.
“We applied the BSRS-5 to measure participants’ general mental health state in the past 1 week. We did not assess participants’ mental health before the COVID-19 pandemic or during a longer period.”
Comment 9
Minor:
1. Line 142 The sex, age, educational… 'The'
2. In all the tables, Covid-19, coronavirus disease 2019.?
3. Line 198. 4. Discussion
Response
Thank you for your reminding.
- We revised tit into “The.” Please refer to line 152.
- We revised it into “COVID-19: Coronavirus disease 2019” in the footnote of all tables.
3. We put it back to the right place. Please refer to line 311.
Reviewer 2 Report
I admire the timeliness of this paper and its implications are useful for policy makers. I agree with the authors that the sample selection via Facebook is a bit worrisome because it is not clear that the Facebook audience is representative of the overall population - here in the US the Facebook audience has been treading older for many years. I would rather the authors mention the possible selection problem when they first discuss their sample than at the end of the paper.
A few minor typographical errors:
Line 142: "Th" should be "The"
Line 151: "controlled fo" should be "controlling for"
Line 165: "vaccination from" should be "vaccination information from"? It seems that there is a word missing in the original submission.
Table 3 - the formatting of the last column is clunky and difficult to read, can the columns for B, SE, and p be narrowed so that the entire CI can be on one line?
Table 5: Notes include ""Regression Analysis4. Discussion." which seems out of place. Also, can the notes include what the dependent variable is for the regressions?
References: Make sure all references have the same font and format.
Author Response
Thank you so much for your comments. As discussed below, we have revised our manuscript based on your suggestions. Please let me know if we need to provide anything else regarding this revision.
Comment 1
I admire the timeliness of this paper and its implications are useful for policy makers. I agree with the authors that the sample selection via Facebook is a bit worrisome because it is not clear that the Facebook audience is representative of the overall population - here in the US the Facebook audience has been treading older for many years. I would rather the authors mention the possible selection problem when they first discuss their sample than at the end of the paper.
Response
Thank you for your comment. We added the discussion about recruiting participants from Facebook in the beginning of Discussion as below. Please refer to line 217-222.
“Before discussion, the limitation of the present study resulted from the method of recruiting participants should be addressed. Although recruiting participants from Facebook is a practical method during the pandemic, the participants might not be representative of the general population [32]. People are not equally motivated to use Facebook [33]. Moreover, a 2018 analysis found that 68.4% of active Facebook users in Taiwan were aged between 18 and 44 [34].”
Comment 2
A few minor typographical errors:
- Line 142: "Th" should be "The"
- Line 151: "controlled fo" should be "controlling for"
- Line 165: "vaccination from" should be "vaccination information from"? It seems that there is a word missing in the original submission.
Response
Thank you for your reminding.
- We revised tit into “The.” Please refer to line 152.
- We revised it into “controlling for.” Please refer to line 161.
- We added “information” here. Please refer to line 175.
Comment 3
Table 3 - the formatting of the last column is clunky and difficult to read, can the columns for B, SE, and p be narrowed so that the entire CI can be on one line?
Response
Thank you for your suggestion. We reorganized Table 3 to make the entire CI on one line. Please refer to Table 3.
Comment 4
Table 5: Notes include ""Regression Analysis4. Discussion." which seems out of place. Also, can the notes include what the dependent variable is for the regressions?
Response
We put it back to the right place. Please refer to line 211. We also included the dependent variables into notes of Tables 3 and 5 by labelling “a: dependent variable.” Please refer to line 188 and 210.
Comment 5
References: Make sure all references have the same font and format.
Response
We rechecked the references and unified them into the same format.
This manuscript is a resubmission of an earlier submission. The following is a list of the peer review reports and author responses from that submission.